# The Preferential Use of Anakinra in Various Settings of FMF: A Review Applied to an Updated Treatment-Related Perspective of the Disease

**DOI:** 10.3390/ijms23073956

**Published:** 2022-04-02

**Authors:** Eitan Giat, Ilan Ben-Zvi, Merav Lidar, Avi Livneh

**Affiliations:** 1FMF Clinic, The Chaim Sheba Medical Center, Tel-Hashomer, Ramat-Gan 5265601, Israel; eitan.giat@gmail.com (E.G.); ilan.benzvi@sheba.health.gov.il (I.B.-Z.); merav.lidar@gmail.com (M.L.); 2Rheumatology Unit, The Chaim Sheba Medical Center, Tel-Hashomer, Ramat-Gan 5265601, Israel; 3The Sackler Faculty of Medicine, Tel-Aviv University, Tel-Aviv 6997801, Israel; 4Medicine F, The Chaim Sheba Medical Center, Tel-Hashomer, Ramat-Gan 5265601, Israel; 5The Talpiot Medical Leadership Program, The Chaim Sheba Medical Center, Tel-Hashomer, Ramat-Gan 5265601, Israel

**Keywords:** familial Mediterranean fever (FMF), anakinra, interleukin 1 blockers, colchicine failure, AA amyloidosis, exertional leg pain, protracted febrile myalgia, chronic renal failure, kidney transplantation, safety

## Abstract

Familial Mediterranean fever (FMF), the most frequent monogenic autoinflammatory disease, is manifested with recurrent and chronic inflammation and amyloid A (AA) amyloidosis, driven by overproduction of interleukin 1 (IL-1) through an activated pyrin inflammasome. Consequently, non-responsiveness to colchicine, the cornerstone of FMF treatment, is nowadays addressed by IL-1- blockers. Each of the two IL-1 blockers currently used in FMF, anakinra and canakinumab, has its own merits for FMF care. Here we focus on anakinra, a recombinant form of the naturally occurring IL-1 receptor antagonist, and explore the literature by using PubMed regarding the utility of anakinra in certain conditions of FMF. Occasionally we enrich published data with our own experience. To facilitate insights to anakinra role, the paper briefs some clinical, genetic, pathogenetic, and management aspects of FMF. The clinical settings of FMF covered in this review include colchicine resistance, AA amyloidosis, renal transplantation, protracted febrile myalgia, on- demand use, leg pain, arthritis, temporary suspension of colchicine, pediatric patients, and pregnancy and lactation. In many of these instances, either because of safety concerns or a necessity for only transient and short-term use, anakinra, due to its short half-life, is the preferred IL-1 blocker.

## 1. Introduction

Familial Mediterranean fever (FMF) is the most frequent monogenic autoinflammatory disease (AID). It is caused by gain-of-function mutations in the Mediterranean fever gene (MEFV) [1], but since the mutations are with low penetrance, and the expression of the gene is dose dependent, the inheritance pattern appears to be autosomal recessive in most patients, rather than dominant. Nevertheless, in 30% of the patients, a single mutated allele is sufficient to express the disease [2,3]. FMF is mostly linked to populations with eastern Mediterranean roots and is characterized by recurrent febrile and painful abdominal, chest, and joint attacks, resulting from acute inflammation of serous membranes lining body cavities [4]. Long periods of inflammation may complicate FMF with reactive amyloid A (AA) amyloidosis [5]. 

Since 1972, colchicine has been the standard treatment of FMF [6], preventing both acute attacks and chronic inflammation in all users, except for a small proportion of FMF patients, refractory to the drug [7]. This gap is now bridged by anti-interleukin (IL)-1 agents, which have greatly impacted FMF treatment and prognosis [8]. Anakinra, a recombinant form of the naturally occurring IL-1 receptor antagonist, was the first IL-1 blocker, and the first to show efficacy in FMF [9], paving the road for other IL-1 blockers. While canakinumab, a human monoclonal anti-IL-1beta antibody, for the convenience of its use, became the preferred IL-1 blocker in FMF, anakinra has its own benefits in certain circumstances of FMF. In recent years, increasing data have been accumulated, confirming anakinra efficacy and safety in the treatment of FMF and leading the European Medicines Agency (EMA) to grant anakinra with an indication for use for FMF [https://www.ema.europa.eu/en/documents/variation-report/kineret-h-c-363-ii-0070-epar-assessment-report-variation_en.pdf (accessed on 29 March 2022)]. However, no guidelines advising on the preferred Il-1 blocker exist. 

This article summarizes current knowledge and our own experience regarding the utility of anakinra in the treatment of FMF and on the background of the most recent advances and treatment dilemmas in FMF. The paper implies that anakinra’s short half-life is an advantage, making it a safer treatment choice in the setting of an increased risk for infection and in clinical conditions where short-term use is required. The use of anakinra rather than canakinumab to control hyperinflammation caused by Corona virus disease -19, just underlines this merit.

## 2. Methods

To study current knowledge on anakinra treatment for FMF, we searched PubMed for all papers which appear under the term “familial Mediterranean fever” crossed with the term “anakinra” through 31 December 2021. Of 191 papers retrieved, we focused on those contributing to current knowledge on the treatment of anakinra in the following situations of FMF: colchicine resistance, AA amyloidosis, renal transplantation, protracted febrile myalgia, on-demand use, leg pain, arthritis, temporary suspension of colchicine, pediatric patients, and pregnancy and lactation. To the data abstracted from the papers, we added our experience with FMF patients receiving anakinra, in instances where our experience was different from published data, or where we could contribute further. To improve the understanding of the role of anakinra we provided clinical, genetic, pathogenetic, and management background on FMF and briefly reviewed the IL-1 cytokine family. 

## 3. Features of FMF

### 3.1. Prevalence

FMF is rare in most parts of the world, but keeping with its Mediterranean origin, it is relatively common among non-Ashkenazi Jews, Turks, Armenians, Arabs, and other ethnic groups living around the Mediterranean basin. Among Armenians, one of five is a heterozygous carrier, and the prevalence of the disease is 1:500 [10]. In some parts of northern Turkey, a prevalence of 0.8% was reported [11]. In Israel, the prevalence is also high and estimated to be around 1:1000 [12,13], but it mostly affects Sephardi, particularly North African Jews [14]. During the past century, the immigration of ethnic groups with high prevalence of FMF to other countries, resulted in a shift of the affected population from around the Mediterranean coasts to inner Europe and to the United States of America (USA), with increased prevalence of FMF in countries such as Germany, France, and the USA [15,16,17]. Recent reports of the occurrence of FMF in countries such as Japan [18], in which FMF was unknown, suggest that FMF may be more prevalent than is currently assumed. 

### 3.2. Genetics and Pathogenesis

#### 3.2.1. MEFV

For over two decades, FMF has been known to be associated with mutations in MEFV, the gene that encodes pyrin [19], a large cytosolic protein (781 amino acids, 86 Kd), that has five functional units. These include a pyrin domain (nucleotides 1–92), a basic leucine zipper transcription factor domain (nucleotides 266–280), a B-box domain (nucleotides 370–412), an α-helical coiled-coil domain (nucleotides 420–440) and a B30.2/SPRY domain (nucleotides 597–776). Each domain is essential for pyrin protein function [20]. MEFV is composed of 10 exons and is expressed mainly in granulocytes, monocytes, and other cells playing a role in FMF biology [19,21]. To date, more than 310 MEFV sequence variants have been recorded in Infevers (https://infevers.umai-montpellier.fr/web/ (accessed on 29 March 2022)), an online registry of known AID mutations [22]. Most of the mutations occur in exon 10, the longest MEFV exon. These include the M694V, V726A, M680I, and M694I mutations, the four most identified in patients with FMF [23]. Most FMF patients carry 2 affected alleles, but mainly those who are homozygous or compound heterozygous to the M694V, M680I, and M694I mutations bear the risk of experiencing a severe disease, with onset at a young age, usually less than 10 years, as well as having colchicine resistance and AA amyloidosis [24,25]. 

#### 3.2.2. Pyrin Function

Pyrin is one of the cytosolic pattern-recognition sensor proteins that recognizes certain pathogen/danger-associated molecular patterns (PAMPs/DAMPs). Pyrin activation results in the assembly of the inflammasome, a multiprotein oligomer that activates caspase-1, a proteolytic enzyme which catalyzes pro-IL-1beta (IL-1β) to the mature form of IL-1β, and initiates an inflammatory response, mainly through the generation and release of IL-1β and IL-18 [26]. Recent evidence demonstrates that pyrin is normally kept in an idle state, caused by its binding to the inhibitory proteins 14-3-3. This binding is actively maintained by the phosphorylation of the pyrin domain. The phosphorylation is mediated by a chain of reactions, starting with constitutive activation of Ras homologous guanosine triphosphatase (Rho GTPase), which induces the action of protein kinase N1 (PKN1) and PKN2, which phosphorylate pyrin. Inactivation of RhoA GTPase by certain bacterial toxins, impedes phosphorylation of the pyrin domain, and disengages the inhibitory proteins from their binding to pyrin. This leads to pyrin activation, production and release of IL-1β, and subsequent inflammation [27]. Based on the pathophysiology in FMF described above, pyrin mutations, particularly in exon 10 (forms the B30.2 domain, a site that interacts with caspase-1), are thought to hamper pyrin phosphorylation and are thereby prone to activate pyrin and cause intermittent and often chronic systemic and local inflammation, manifested with the FMF phenotype [20,27].

Of note, different insights to the pathogenesis of FMF, unrelated to the pyrin phosphorylation, have been forwarded, based on observations made by other workers. One major step in the inflammasome activation is the binding of pyrin to another large protein called apoptosis-associated speck-like protein containing CARD (ASC). This step is dependent on prior polarization of cellular microtubules, required for ASC clustering. It has been shown that in FMF the assembly of the FMF inflammasome is independent of microtubule polarization. Thus, it is thought that MEFV mutations cause pyrin priming, which allows its binding to ASC and FMF inflammasome activation without the microtubule polarization stage [28].

#### 3.2.3. Modifier Genes

Other genes, not cloned yet, presumably encoding for proteins upstream or downstream of the metabolic pathway of pyrin, may be associated with the occurrence of FMF. This is exhibited by cases of genetic-negative FMF, comprising 20% of the FMF population [29]. In addition, phenotype–genotype correlation in FMF is far from being optimal with differences in FMF manifestations even among monozygotic twins [30]. Thus, certain variations, occurring in several inflammatory or innate immunity-related genes, were screened to find predisposition to FMF, or an effect on the phenotype of the disease. Indeed, increased propensity to develop FMF (determined based on higher allele frequency) was found for serum amyloid A (SAA)1.3 and for human leukocyte antigen (HLA)B39:01 in Japanese populations [31,32], The effect of the HLAB39 locus was more pronounced in patients bearing non-classical MEFV mutations. However, by using haplotype analysis, earlier results excluded linkage between SAA whole gene cluster and FMF in Israeli and Armenian populations [33,34]. Similarly, HLA antigen analyses appeared comparable in FMF patients and control subjects in Israeli and mixed, mostly Armenian populations [35,36]. These conflicting reports may result from methodological and ethnic variations. Regarding the expression of FMF, several gene polymorphisms or HLA-loci were found to variably impact the severity, age of onset, response to colchicine, and development of amyloidosis. These include some alleles of the major histocompatibility complex class I chain related A (MICA), SAA1α/α genotype, particular nucleotide binding and oligomerization domain (NOD)-2 mutations, as well as the pR753G mutation of toll-like receptor-2 [37,38,39,40]. These data only evince the complexity of the pathogenesis of FMF. Of note, large numbers of inflammatory gene polymorphisms failed to show an association with FMF. Of note are the IL-1β C511T and IL-6 G174C variations [31,41]. 

### 3.3. Manifestations

#### 3.3.1. FMF Short Attacks

FMF typically presents with recurrent bouts of fever, usually with serositis or skin rash, that last for up to four days, and resolves spontaneously. Attacks occur irregularly, with a frequency spanning from once in several days to several months, or even less frequently [14]. Often an attack is proceeded by a prodrome, with one or two of a range of manifestations, predicting its emergence [42]. Between the flares, the patients are free of symptoms of the acute attacks, but chronic manifestations may exist (see Section 3.3.2). Emotional stress, fatigue, surgery, menstruation, vigorous exercise, physical trauma, and cold exposure may trigger an attack, but some of these precipitants are doubtful [43,44]. The approximate age of onset in children is 2.6 years (range 1.1 to 5.3), with a delay in diagnosis of 3.3 years (range 2.5 to 4.3). Diagnostic delay in adults is longer, exceeding a mean of 10 years [45,46,47,48]. 

The most common attacks are abdominal, affecting about 90% of patients [45,46]. These attacks often present with agonizing peritonitis, mimicking acute abdomen of any cause, occasionally resulting in unnecessary abdominal operations (e.g., appendectomy) [14]. Attacks of painful monoarthritis, mostly of the large joints of the legs, which often become swollen and very red, mimicking septic arthritis, are less frequent, occurring in about 30% of the cases [45,46]. Not uncommonly, arthritis attacks may be precipitated by effort or trauma. Chest attacks, characterized by an acute one-sided pleuritis or pericarditis, with painful inspiration and occasionally diminished breathing sounds, affects 15 to 54% of the patients [45,47]. 

The skin may be affected with attacks of tender, hot, swollen, sharply bordered red eruption, usually between the knee and ankle, entitled erysipelas-like erythema (ELE) [14,49]. This type of attacks is less common (5%), and should be discriminated from periarthritis, a much more frequent erythematous discoloration (30%), involving the skin overlying inflamed joints during arthritis attacks, particularly in the ankle and dorsum of the foot areas. While ELE is highly specific to FMF, periarthritis is not, concurring with FMF but also with infectious, sarcoidosis, and reactive arthritides [50]. Finally, attacks of fever alone, without involvement of any of the above sites, but occasionally inflicted with constitutional manifestations (e.g., headache, generalized muscle and joint pain) are also characteristic of FMF [14,51]. Fever attacks are more common in children and may form the first manifestation of the disease [24]. 

#### 3.3.2. Other Manifestations of FMF

In addition to the spontaneous short-term inflammatory spells, FMF phenotype includes chronic, or protracted, or inducible manifestations, the most common of which is symmetrical leg pain, particularly involving the calves, but also or only the thighs, knees, ankles, and feet [4]. This very characteristic manifestation of FMF, usually develops during physical activity, notably prolonged standing, and therefore is termed exertional leg pain (ELP). It is more common in patients homozygous to the M694V mutation, and is caused at least in part by enthesitis and tenosynovitis [52,53]. A severe form of leg pain may progress to arthritis involving both ankles simultaneously (contrasting spontaneous episodic ankle arthritis which is one sided), and/or manifest continuously without physical stimulation. Often ELP with or without elevated inflammatory markers form the only manifestation of FMF, particularly in carriers of only one MEFV mutation.

Protracted febrile myalgia (PFM) is another prolonged disease manifestation (1–3 months), typical of FMF. It is characterized by excruciating and debilitating myalgia, fever, occasionally spotted skin rash, and infrequent abdominal pain, without peritoneal irritation. Inflammatory markers are exceedingly high [54]. It is thought to result from vasculitis. In general, vasculitides are more common in patients with FMF than in the general population, most commonly presenting as Henoch Schönlein purpura in 3 to 11% of patients [47], poly-arteritis nodosa [55], and Behcet’s disease [56,57,58]. In contrast to other vasculitides, PFM is specific to and diagnostic of FMF. Chronic inflammation of serous membrane (e.g., chronic pericarditis or arthritis), were also reported, but they are extremely rare [59,60]. FMF patients are also afflicted with other inflammatory disorders, at a higher rate compared to the general population, notably human leukocyte antigen B27-negative spondyloarthritis [57,61] and inflammatory bowel diseases [62,63,64].

Undoubtedly, reactive AA amyloidosis is the most dreadful chronic manifestation of FMF. In the pre-colchicine era, it was common in FMF, affecting up to 60% of patients [65]. Usually, manifestations of amyloid nephropathy govern the picture, beginning with asymptomatic proteinuria and gradually evolving through a nephrotic phase to end stage renal disease over 2–20 years. Currently, the prevalence of amyloidosis has dropped significantly, resulting from colchicine treatment and perhaps environmental and epigenetic factors [66]. Yet it remains a serious therapeutic challenge, with a five-year mortality rate of 50% [67]. Amyloidosis is further discussed in Section 7.4. 

There is no pathognomonic laboratory test for FMF. However, typically during FMF attacks there is a prompt and marked rise in the acute-phase reactants, including erythrocyte sedimentation rate (ESR), C-reactive protein (CRP), SAA, and fibrinogen [68]. In about 30% of patients, these inflammatory markers are elevated continuously, although to a lesser extent, implying chronic inflammation [69].

### 3.4. Diagnosis

In the absence of a particular, widely available and validated, distinctive laboratory marker, FMF is diagnosed by its typical clinical picture. Results of genetic testing may support the diagnosis, if two MEFV mutations are detected, but not exclude it, if only one or none of the mutations are found [3,29,70]. Moreover, individuals bearing two MEFV mutations, but free of symptoms, cannot be diagnosed with FMF. Based on screenings of the general population, the number of unaffected subjects who bear two MEFV mutations exceeds by far the number of FMF patients [71,72]. Various diagnostic criteria have been established, including the Tel Hashomer criteria [45,73,74], which is the most widely used, due to its simplicity, accuracy and compliance with the highly specific presentation of FMF. In a recent study, by using a large international registry of autoinflammatory diseases (Eurofever), two sets of valid, evidence-based classification criteria (one clinical–genetic and one purely clinical) have been established to differentiate between the five most common autoinflammatory syndromes: FMF, cryopyrin-associated periodic syndromes (CAPS), mevalonate kinase deficiency (MKD), tumor necrosis factor receptor-associated periodic fever syndrome (TRAPS), and periodic fever, aphthosis, pharyngitis, and adenitis (PFAPA). Because for building classification criteria specificity takes precedence over sensitivity, atypical cases may be missed by this tool; therefore the authors advised against using their classification criteria for FMF diagnosis [75].

## 4. Treatment of FMF

### 4.1. Colchicine

#### 4.1.1. Mechanism of Action

Colchicine has been used for the treatment of FMF since 1973, based on an observation made by Goldfinger SE [76] and two controlled trials [6,77]. Colchicine is an alkaloid with inhibitory effects on multiple cellular functions, including microtubule assembly, cell adhesion, and inflammasome activation [78]. Inhibition of microtubule assembly enhances the release of guanine-nucleotide-exchange factor -H1 (a known regulator of RhoA proteins) from damaged (depolymerized) microtubules [79], which induces RhoA activation, thereby promoting phosphorylation of pyrin, which in turn leads to inhibition of pyrin inflammasome and IL-1β production, as detailed above, in the pathogenesis Section 3.2.2 [27]. Another study has demonstrated that colchicine suppresses ASC aggregation and binding to pyrin, thereby disrupting pyrin inflammasome activation through a mechanism completely unrelated to pyrin phosphorylation [80]. 

#### 4.1.2. Efficacy and Safety of Colchicine in FMF

Continuous administration of colchicine prevents FMF attacks in about 65% of patients and induces partial remission in a further 25% [81]. In addition, appropriate use of colchicine prevents amyloidosis, which is the primary cause of premature death in FMF [82]. Colchicine dose ranges between 1 and 3 mg per day, as determined individually by its efficacy in preventing attacks and chronic inflammation, on the one hand, and tolerance of its use and other safety issues on the other hand. According to the recently published European Alliance of Associations for Rheumatology (EULAR) guidelines [83], colchicine should be started as soon as a clinical diagnosis is made. 

There seems to be a hierarchy in the responsiveness of FMF manifestations to colchicine treatment. Fever and abdominal, chest, and scrotal attacks are the most susceptible to colchicine’s effects. Joint attacks and leg pain respond less favorably. The least affected by colchicine treatment is the state of chronic inflammation with elevated acute phase reactants, which may remain the last FMF manifestation in some FMF patients even if professionally managed with the maximally tolerated dose of colchicine [personal experience].

Although colchicine is safe and efficacious in most patients, some patients suffer adverse events, of which diarrhea and abdominal soreness is the most common [84,85]. Rarely neutropenia, myopathy, neuropathy, and hair loss may take place. Whether colchicine induces liver toxicity in an ordinary dose is still debated, as most liver enzyme abnormalities in FMF treated with colchicine are thought to result from FMF-related inflammation [86].

#### 4.1.3. Colchicine Resistance and Intolerance

Approximately 10% of patients are refractory to colchicine treatment, as determined by a failure to reduce attack frequency to less than 1 per month, despite good adherence to a maximal tolerated dose [7,83]. A more challenging definition would require colchicine to reduce attack frequency (of any type) to less than 1 attack per three months, completely abrogate exertional leg pain, and normalize inflammatory markers in attack-free periods. Although most FMF-related amyloidosis results from inappropriate treatment, it should be included in the definition of colchicine failure and managed as such. This more stringent definition undoubtedly will increase the fraction of patients considered unresponsive to the drug to an estimated figure of 20%. Intolerance to colchicine, defined as an inability to mount colchicine dose to a prophylactic level due to adverse events, usually diarrhea, is also included in the above estimate of the frequency of colchicine treatment failure. In a recent publication of an international consortium most of these insights have been approved [87]. 

The reasons for treatment failure in patients with colchicine-resistant FMF are complex and not fully understood. An early study [7] comparing colchicine responders to non-responders who shared similar ethnic, demographic, colchicine-treatment and FMF-genetic features, found that plasma and polymorphonuclear colchicine concentrations were comparable in both groups. However, colchicine concentration in the mononuclear cells was significantly lower (half the level) in non-responders, suggesting that colchicine treatment failure might be related to a genetic defect in a metabolic pathway controlling colchicine influx or efflux in this cell population [7].

Indeed, various polymorphisms of the drug transporter adenosine triphosphate-binding cassette subfamily B member 1/multidrug resistance protein 1, considered important for colchicine cellular transport, were demonstrated to be more common in colchicine refractory FMF [88]. However, the effect of these changes on colchicine cellular content was not investigated, precluding the implication of these findings in the mechanism of colchicine failure. Other associations with colchicine resistance include low concentrations of serum 25-hydroxy vitamin D [89], and blood group O [90], the pathogenicity of which is obscured.

Because of colchicine resistance, FMF patients continue to endure severe serositis attacks, confining them to bed, impeding their school and work opportunities and success, subjecting them to a poor quality of life, and increasing their risk of dying from amyloidosis or developing other conditions related to chronic inflammation [87,91,92]. Thus, many attempts were made over the years to change their fate. In Israel, intravenous colchicine, in a dose of 1 mg/week in addition to oral colchicine, is still being used successfully for colchicine treatment failure, without a significant toll in safety [93]. Unfortunately, this treatment is not available outside Israel, because it was banned by the American Food and Drug Administration (FDA) and taken off the markets, due to improper use in conditions unrelated to FMF (e.g., lower back pain), resulting in fatal toxicity [94]. Other therapeutic alternatives for non-responders that have been shown to be beneficial, include thalidomide [95], interferon alpha [96], various tumor necrosis factor (TNF) blockers [97,98], interleukin-6 blockers [99], tofacitinib [100], and selective serotonin reuptake inhibitors [101], all of which are not adequately scientifically based, limited by insufficient or transient effects, safety concerns, unavailability, and high cost, and none of which are currently approved by a regulatory body. 

More recently and following the finding that FMF activity is mediated by the proinflammatory cytokine IL-1β, IL-1 blockers have been proposed as a possible treatment for colchicine-resistant FMF. Accordingly, the IL-1β antibody canakinumab was the first IL-1 blocker approved for treatment of FMF. A little later, anakinra, the first synthesized IL-1 blocker and the first to be released to the market for treatment of a variety of inflammatory disorders, received an indication for FMF by EMA.

### 4.2. Canakinumab

Canakinumab is the first IL-1 blocker approved for FMF [102]. It is a human monoclonal antibody specifically targeted at IL-1β. It does not react with interleukin-1 alpha [103]. It has been approved by the FDA and the EMA for the treatment of FMF and several other inflammatory disorders. In a major difference from anakinra, which is given daily (see Section 6.1), canakinumab is usually administered every 4–8 weeks.

The efficacy and safety of canakinumab for colchicine resistant FMF was demonstrated in the CLUSTER study, a randomized, placebo-controlled trial (RCT), which included 63 patients with colchicine resistant FMF [104]. At the end of the study, a complete response was achieved in 71% of the patients. The most frequently reported adverse events were infections, occurring at a rate of 6.6 per 100 patient years. A recent systematic review, assessing the published outcome of canakinumab treatment, likewise confirmed its favorable effect [105]. These data further emphasize the important role of anti-IL-1 therapy in the treatment of FMF and of canakinumab as one of its representatives.

### 4.3. Anakinra

Anakinra, its structure, features, and indications, including use in FMF, and its preferential use in certain conditions of FMF is described separately and extensively in the next chapter (Section 5).

## 5. IL-1 Cytokines, Receptors, and Antagonists

### 5.1. IL-1α and IL-1 β

IL-1α and IL-1β belong to the IL-1 cytokine family that plays a central role in the regulation of the immune and inflammatory responses [106,107,108]. The genes of both were cloned in the 1980s from a macrophage-complementary deoxyribonucleic acid (cDNA) library [109], and the resulting cytokines were found to share a common receptor, IL-1 receptor (IL-1R) [110]. Like all members of the IL-1 cytokine family, they are first synthesized by cells in a precursor form. 

IL-1α is a double-function protein, acting both as a cytokine, and as a transcription factor. Its precursor is present constitutively in epithelial cells, keratinocytes, and fibroblasts (but not in myeloid cells), as a cytosolic or membrane-bound protein [111,112], and is processed to its mature 17-Kd form by the Ca^2+^-activated protease calpain. Because of the abundance of calpain inhibitors in most cells, IL-1α is processed minimally and therefore its inflammatory role through these cells and in general is limited. During inflammation IL-1α is synthesized de novo and actively secreted almost exclusively from myeloid cells [113] and acts as a DAMP to trigger an immune response [106,114].

IL-1β is expressed in inflammatory cells as an inactive precursor protein, only after the exposure of the immune cells to certain alarmins (e.g., lipopolysaccharide). An additional stimulus by alarmins leads to the assembly of the inflammasome and the subsequent activation of caspase-1, which in turn cleaves the pre-IL-1β to its active form and mediates its secretion. IL-1β is a potent pro-inflammatory cytokine, which upregulates the expression of many other inflammatory cytokines, including IL-1α [115]. IL-1β-deficient mice did not manifest acute phase response, or elevated temperature, upon subcutaneous injection of turpentine (which solely leads to IL-1 and IL-6 activation). In contrast, administration of lipopolysaccharide, which activates many more cytokines, did provoke an inflammatory response comparable to that of wild-type mice [116], attesting to the complexity of triggering inflammation, which may bypass IL-1β. 

### 5.2. IL-1 Receptor (IL-1R)

The IL-1R family consists of 11 receptors with roles in promoting or inhibiting inflammation and immune reactions, notably innate immune reactions [117]. They all have an intracellular toll/interleukin-1R (TIR) domain, which may interact with cytosolic adaptor proteins such as myeloid differentiation factor 88 (MyD88). Downstream signaling is attained through recruitment of other TIR domain-containing proteins and homotypic binding of the adaptor proteins to IL-1R-associated kinases. This signal transduction pathway leads to the release and nuclear localization of the transcription factor nuclear factor kappa light chain enhancer of activated B cells (NF-κB), a potent regulator targeting immune and inflammatory genes, such as IL-1, IL-6, TNF-α, and many more [118]. The intracellular TIR domain and its signal transduction pathways are also found in toll-like receptors, an important group of receptors that recognize PAMPs, highlighting their key role in the innate immune system. 

The extracellular part of most IL-1R members consists of three immunoglobulin (Ig)-like domains, which bind specifically to the cytokine. Both IL-1α and IL-1β bind to IL-1R1. IL-1 signaling requires the formation of a hetero-trimeric complex, containing the ligand (i.e., the cytokine), the receptor and the co-receptor IL-1R3. The binding of IL-1β or IL-1α to IL-1R1 causes a conformational change in IL-1R1, which allows the IL-1R3 binding. The formation of the heterotrimeric complex brings the intracellular domains of IL-1R1 and IL-1R3 into proximity, which enables the TIR domains to recruit the adaptor protein MYD88 and start the signal transduction. Contrary to IL-1R1 and IL-1R3, IL-1R2 has no cytoplasmic TIR domain. It binds to IL-1β and forms a hetero-trimeric complex with the co-receptor IL-1R3, but due the absence of the TIR domain, it does not induce a downstream signal. Therefore, it is considered a decoy receptor that serves as another mechanism to inhibit the inflammatory effects of IL-1β. 

### 5.3. IL-1R Antagonist (IL-1RA)

IL-1RA was first described in the 1980s in the blood and urine of patients with JIA as an inhibitory protein for IL-1 activity. It binds to the IL-1R and competitively interferes with the binding of IL-1α and IL-1β to IL-1R1. When IL-1RA binds to IL-1R1, a conformational change occurs that prevents the binding of the co-receptor IL-1R3 and thereby the proximation of the TIR domains and the subsequent signal transduction. IL-1RA binds to IL-1R1 with a higher affinity than either IL-1α or IL-1β, making it a highly effective receptor antagonist. Deficiency in IL-RA due to mutations in IL-1RN, the gene encoding IL-RA, results in deficiency of the IL-1RA (DIRA). This is an autosomal-recessive, autoinflammatory disease characterized by sterile multifocal osteomyelitis, periostitis, and pustulosis [119], underlining the critical role of IL-1RA in regulating inflammation. 

## 6. Anakinra

### 6.1. Anakinra’s Features

Anakinra is a recombinant, non-glycosylated form of human IL-1RA. It consists of 153 amino acids and has a molecular weight of 17.3 Kd. It is produced by recombinant DNA technology, by using an *E. coli* bacterial expression system, and differs from the native IL-1RA by the addition of a single methionine residue at its amino terminus [120]. Both IL-1RA and anakinra bind to the IL-1R in the same manner and interfere with the binding of IL-1α and IL-1β, thereby blocking their activity.

Anakinra is administered subcutaneously (SC) in a daily injection containing 100 mg/0.67 mL solution. Bioavailability of anakinra after a SC injection is around 95%, with the maximum plasma concentrations found 3–7 h after administration. The terminal half-life is between 4–6 h. Anakinra clearance is affected by the glomerular filtration rate (GFR); mean plasma clearance in subjects with mild (GFR of 50–80 mL/min) and moderate (GFR of 30–49 mL/min) renal insufficiency was reduced by 16% and 50%, respectively. In severe renal insufficiency (GFR of 15 to <30 mL/min) and end-stage renal disease, mean plasma clearance declines by 70% and 75%, respectively. Less than 2.5% of the administered dose of anakinra is removed by hemodialysis or continuous ambulatory peritoneal dialysis. Therefore, the manufacturer recommends administering anakinra every other day to patients with severe renal insufficiency or end-stage renal disease (GFR < 30 mL/min). Weight, gender and age are not significant factors for plasma clearance [https://www.accessdata.fda.gov/drugsatfda_docs/label/2016/103950s5175lbl.pdf (accessed on 29 March 2022)].

### 6.2. Anakinra in Diseases Other Than FMF

The beneficial effects of anakinra have been demonstrated in the treatment of a wide range of disorders, including auto-inflammatory diseases such as CAPS, TRAPS, or MKD/ HIDS [121], wherein the pathogenesis is associated with an excessive IL-1 expression. In CAPS (an entity consisting of familial cold autoinflammatory syndrome, Muckle–Wells syndrome and neonatal-onset multisystem inflammatory disease), anakinra has been shown to prevent and even reverse the manifestations, such as fever, rash, arthritis, fatigue, conjunctivitis, headache, papilledema, and mental and hearing impairment. It even stopped the evolution of hydrocephalus. These favorable clinical outcomes were associated with normalization of inflammatory markers, blood cell counts and cerebrospinal fluid constituents, altogether leading to improved well-being and quality of life [121,122,123,124,125,126]. In TRAPS, an autoinflammatory disorder caused by mutations in TNF receptor type 1, anakinra has been effective in treating refractory cases [127], suggesting an important role for IL-1 in the inflammatory process created by a defect in a TNF-mediated disease. Anakinra is also effective in non-monogenic auto-inflammatory disorders such as PFAPA [128], adult-onset Still’s disease (AOSD) [129], systemic onset JIA (SOJIA) [130], and Schnitzler’s syndrome [131], an inflammatory disorder associated with plasma cell dyscrasia and manifested with chronic urticaria and fever. Other diseases that were shown to respond to anakinra include severe seizure disorder, termed febrile infection-related epilepsy [132], pre-myeloma [108], gout [133], chronic and recurrent pericarditis [134], Cogan’s syndrome [135], and others [136]. 

## 7. Anakinra in FMF

### 7.1. Preface

In recent years, anakinra has emerged as an important tool for the treatment of FMF, its complications, and related conditions. The main situations for which anakinra is required is colchicine-resistant FMF and intolerance to colchicine (defined in Section 4.1.3), leading to recurrent attacks, uncontrolled leg pain and chronic inflammation with amyloidosis in its extreme (detailed in Section 3.3.1 and Section 3.3.2). There are other conditions unanswered by colchicine therapy, including acute FMF attacks at their outburst, protracted febrile myalgia (Section 3.3.2), and more. In all, anakinra has been tested, including during childhood and pregnancy. Table 1 briefly displays various conditions or scenarios in which anakinra has been used successfully, and the experience gained in these situations is more broadly addressed below. 

The short half-life of anakinra provides several advantages, over preparations with longer body persistence. This is true for safety concerns in immunocompromised subjects, where short-lived immunosuppressants confer lower risk in case of infection. The short half-life of anakinra also makes it the preferred IL-1 blocker for short-term use (several days) and for scenarios with diagnostic uncertainty (FMF vs. simulating condition). Table 2 presents these conditions, in which anakinra is the preferred IL-1 blocker. Of note, the listed conditions reflect our practice and logical inference and should be viewed as expert opinion. The preferential use of anakinra will be further discussed in the following sections.

### 7.2. Anakinra as Add-On Therapy to Colchicine

At present, anakinra, as well as other IL-1 blockers, should be administered as an add-on treatment, given in addition to colchicine, and when applicable, colchicine should be continued at a maximally tolerated dose. This is because colchicine is still the only medication known to prevent amyloidosis of FMF. In a study by Zemer et al. [137], amyloidosis was developed in four of 1000 patients treated with colchicine at a dose preventing attacks or at a dose of at least 2 mgs/day, if attacks continued. Compared to historical data from the pre-colchicine era, this finding suggested that the risk of developing amyloidosis in FMF patients managed appropriately with colchicine was extremely low, thereby establishing its role as the mainstay in the prevention of amyloidosis of FMF. Thus, until anakinra is found to act comparably, colchicine should not be discontinued (see more on anakinra monotherapy in Section 7.10).

### 7.3. Anakinra for Colchicine Resistant FMF

#### 7.3.1. Clinical Trials

Currently, there is only one prospective, double-blind RCT from 2017, which has demonstrated the superiority of anakinra over placebo in FMF [138]. This study, performed by our team, was an investigator-initiated single-center trial. It enrolled 25 adult patients (ages 38.4 ± 10 and 36.1 ± 12.4 years in 12 anakinra vs. 13 control patients), carrying at least two MEFV mutations (mostly M694V), who experienced at least one attack per month (4.6 ± 4.3 vs. 5 ± 2.5), in any of four FMF sites of attack, despite a maximal tolerated dose of colchicine (2.2 ± 0.8 vs. 2.1 ± 0.5 mg /day). Over a treatment period of four months, anakinra appeared to be better than placebo in all studied aspects including: the mean number of attacks per patient per month (1.7 ± 1.7 vs. 3.5 ± 1.9, *p* = 0.037), the number of patients with <1 attack per month (6 vs. none, *p* = 0.005), actual levels of last CRP (3.9 ± 3.6 vs. 19.9 ± 1.8 mg/L) and SAA (11.1 ± 19.1 vs. 110.3 ± 131 mg/L), and number of patients achieving the modified FMF50 improvement criterion (10 of 12 vs. 4 of 13, *p* < 0.008). These successful results were not associated with loss of safety, as the number, rate, and severity of the adverse events in anakinra and placebo groups were similar. 

#### 7.3.2. Case Series

Alongside this RCT, case reports and series of more than 300 patients, receiving anakinra for a duration of up to approximately three years have also been published. Although the treatment period was much longer and the population much larger, than in the RCT, the results were comparable. Anakinra has been shown to reduce attack rate and severity, decrease the levels of inflammatory markers, and achieve improvement in composite measures, that included patient and physician global assessments such as FMF50 or global assessment score [139,140,141,142,143]. Usually, the toll of safety was minimal, mainly consisting of injection-site reactions, as discussed separately in Section 7.13.

### 7.4. Anakinra and FMF-Amyloidosis

AA Amyloidosis is a lethal complication of FMF, resulting from long-standing inflammation and IL-1 induced, excessive production of SAA. This complication leads to massive proteinuria, renal failure, severe diarrhea, malabsorption, dysfunction of other organs, and death [144]. Important risk factors are persistent high levels of SAA and other inflammatory cytokines, homozygosity for the M694V MEFV mutation, and family history of FMF-related amyloidosis [145]. In the pre-colchicine era, amyloidosis affected a considerable proportion of FMF patients. Colchicine treatment markedly reduced the prevalence of amyloidosis, but the risk to develop amyloidosis persists, particularly in colchicine refractory, untreated, or improperly treated patients. At present, amyloid nephropathy of FMF is still considered irreversible, as virtually all affected patients eventually reach end-stage kidney disease (see on FMF-amyloidosis also in Section 3.3.2). 

Shortly after the introduction of anakinra for the treatment of FMF, increasing evidence has begun to accumulate, suggesting that anakinra may favorably affect FMF-amyloidosis. Based on published data, it appears that anakinra may lead to stabilization [143,146,147], regression [139,142,147,148], and normalization of urinary protein levels, even in patients with nephrotic range proteinuria [139,149]. Pitifully, this beneficial effect is limited to early stages of amyloidosis, before significant renal failure develops [146,147,149]. However, the exact cut-off of GFR values, under which kidney function is unlikely to ameliorate, have not been established. Moreover, it remains still to be determined whether anakinra monotherapy (without colchicine) will prevent amyloidosis, in unaffected colchicine refractory patients. 

Unfortunately, it is still speculative to view reversal of proteinuria, as indicative of regression of amyloid burden. At present, no histological conformation for such an analogy has been obtained. At most, regression or stabilization of proteinuria should be interpreted as resulting from a decline in AA deposition and its associated toxic effects [150], or from a decrease in IL-1 associated kidney inflammation [151]. 

### 7.5. Renal Transplantation and Anakinra

FMF-amyloidosis patients undergoing renal transplantation have increased mortality, morbidity, and chronic graft rejection compared to non-FMF patients undergoing renal transplantation [152]. Whether anakinra may change this fate and reduce all cause- or transplant-associated morbidity and mortality is currently unknown. Particularly, insufficient data exist on whether anakinra may increase graft survival, and delay or even prevent development of graft amyloidosis. As management with IL-1 blockers in FMF-amyloidosis is currently the standard of care, it would be unethical to conduct placebo-controlled trials. Observational analyses compared to historical data are therefore awaited to resolve these uncertainties.

Nevertheless, from the available, uncontrolled and very scant data, it implies that anakinra is comparably effective in transplanted and un-transplanted FMF-amyloidosis patients regarding suppression of FMF attacks, stabilization or reversal of proteinuria, and amelioration of renal function [147,149,153,154]. Interestingly, reversal of 18 g proteinuria to almost normal levels and normalization of serum SAA was achieved with anakinra in an FMF-amyloidosis patient who inadvertently received normally functioning amyloid laden kidney [155]. One study argues for the peri- and post-transplantation use of anakinra to counteract the direct deleterious effect of IL-1 on renal function [156]. Yet, another study raises concerns about a possible detrimental role of anakinra in inducing post transplantation allograft rejection [157]. 

We prefer treatment with anakinra over canakinumab in FMF-amyloidosis patients undergoing renal transplantation. We think that for immunocompromised patients, who already receive prednisone, tacrolimus, and mycophenolate to prevent rejection, the short half-life of anakinra provides a safety advantage, in case of intercurrent infection. This approach is not yet evidence based. At present, our unpublished experience is limited to six transplanted patients with four exposed to anakinra for a mean duration of 41 months and four to canakinumab (mean 21 months). Two of these patients received both drugs. With respect to efficacy, both IL-1 blockers were equally competent in preventing attacks and protecting against renal failure and proteinuria. Over the treatment period, two severe infections occurred. One was a life-threatening influenza infection, which led to a switch from canakinumab to anakinra, and the other one was a hepatitis B virus infection, which led to a 4-month suspension of anakinra treatment. Anakinra was resumed after the infection was controlled with entecavir. We feel that our safety policy favoring anakinra over canakinumab for kidney-transplanted patients gains support from these episodes. 

### 7.6. Protracted Febrile Myalgia

PFM is a rare but severe and extremely painful manifestation of FMF (see Section 3.3.2). Colchicine is ineffective, and treatment consists of prolonged (at least six weeks) use of high doses (1 mg/kg) of prednisone. Two doses of anakinra have led to recovery of PFM with a return of ESR and CRP back to normal levels within a week [158]. Anakinra was also effective in PFM patients with prednisone failure or partial response to it [159,160]. These reports, though preliminary, may imply a distinct pathogenesis for this rare disorder, separating it from other types of FMF-associated disorders such as vasculitides (e.g., polyarteritis nodosa) or arthritides (e.g., sacroiliitis), which fail to respond favorably to both colchicine and anti-IL-1 blockers. Of note, our experience with anakinra in a patient with severe and debilitating PFM, who failed prednisone was not favorable. She received SC anakinra for one month and continued to suffer of severe muscle pain for three months, although following anakinra treatment, the pain was less severe, and she could ambulate (unpublished data).

### 7.7. FMF—Related Pericarditis

Both anakinra and canakinumab have been used successfully in idiopathic pericarditis [161]. Nevertheless, there are case reports of patients with recurrent idiopathic pericarditis that failed to respond to canakinumab and recovered while on anakinra [162]. We are aware of an adult FMF patient, in whom anakinra efficiently prevented pericarditis after a failure to respond to canakinumab (unpublished data). In contrast, two patients recently presented to us with chronic FMF-related pericarditis, a condition rarely occurring in FMF (see Section 3.3.2), are currently successfully treated with canakinumab. More research is required to determine the significance of these observations.

### 7.8. On-Demand Use of Anakinra

On-demand use of anakinra is devised as a treatment strategy for acute attacks, which may occur in either of the following: (a) colchicine-resistant FMF, for which continuous treatment with IL-1 blockers was declined and therefore bouts continue, and (b) colchicine-responsive FMF, for which current colchicine dose is insufficient. Because there is no treatment to arrest acute attacks, the management paradigm is aimed only at alleviating pain with analgesics and opioids. On-demand treatment of acute attacks with anakinra is proposed to fill this gap. The short half-life of anakinra is an advantage for the treatment of acute attacks as it is safer than long-term preparations under the circumstances of diagnostic uncertainty, characterizing acute FMF attacks. Also, its short half-life and much lower cost allow flexibility in dose adjustment more readily (e.g., deciding on dose increment). 

Initial experience with the on-demand approach is encouraging. In a cohort of 15 patients with colchicine-refractory FMF, on-demand administration of anakinra, mainly during the prodrome, resulted in a significant reduction of the severity, duration, and frequency of the attacks, compared to the pretreatment values in the same patients, though elevated CRP levels, remained unchanged [163]. Thus, implementation of the on-demand policy might be a reasonable solution for certain, carefully selected patients.

### 7.9. Anakinra and Leg Pain and Arthritis

Exertional leg pain, even nowadays, remains a poorly understood manifestation of FMF (Section 3.3.2). As noted before (Section 4.1.2), arthritis attacks and leg pain on exertion respond less favorably to colchicine, often remaining the only enduring manifestations in FMF patients on high-dose colchicine, rendering them colchicine-resistant (Section 3.3.1). This suggests that colchicine is either a weak IL-1 blocker, or these manifestations are mediated by other mechanisms. Experience with IL-1 blockers, initially observed in anakinra-treated patients, illuminates this quandary. In the anakinra RCT (Section 7.3.1), as well as in other studies, anakinra was found to control arthritis spells even better than the attacks in other sites [138,140,142,147]. Such a favorable response however, was not noted in FMF patients with concurrent spondyloarthritis, probably because other cytokines, such as TNF are also involved in the arthritis of these patients [142]. 

Similarly, debilitating exertional leg pain, adversely impacting quality of life, often remains the only symptom in FMF patients who are otherwise stable under colchicine treatment. As with arthritis, this symptom is effectively addressed by anakinra [138]. The relief provided by anakinra allows patients, who virtually could hardly stand longer than a few moments, to perform much better physically, socially, at home, and most importantly at work. The conclusion drawn from this favorable effect of anakinra is that both arthritis attacks and leg pain on exertion, despite being less controllable by colchicine, are utterly mediated by the IL-1 cytokine. 

### 7.10. Transient and Permanent Replacement of Colchicine by Anakinra

Due to its short half-life, anakinra is an ideal drug to replace colchicine, if temporary suspension of colchicine treatment is required. This may occur under certain circumstances, including life-threatening conditions instituted by continuous administration of colchicine (e.g., colchicine intoxication, and other conditions listed in category 2 of Table 2). Our experience with this approach is limited to two patients with colchicine intoxication. But the strategy was successful, and while on anakinra, the patients had no FMF attacks, despite cessation of colchicine treatment (unpublished observation). Substitution of colchicine for a short time may also be required when its use is limited by several other nonthreatening conditions (e.g., due to abdominal operation and other conditions listed in category 3 of Table 2). Permanent replacement of colchicine by IL-1 blockers is usually due to intolerance or allergy to colchicine. In these conditions, using anakinra has no advantage over switching to canakinumab, unless the patients are immunocompromised or suffer of recurrent infections, as discussed above (Section 7.5). Our experience with anakinra and canakinumab under these circumstances is favorable, yet is limited to five patients. The beneficial use of anakinra as monotherapy in the above conditions, should not encourage colchicine withdrawal in colchicine-resistant FMF. However, although this is currently not advised (Section 7.2), we do try to gradually reduce the colchicine dose, under the auspices of anakinra, with careful clinical (looking for attacks or leg pain) and laboratory (looking for a rise in inflammatory markers) monitoring. The experience gained so far with this practice is currently insufficient to draw conclusions. 

**Table 2 ijms-23-03956-t002:** Conditions for which anakinra is preferentially advised.

Reason for Preferential Use of Anakinra	Condition	Category
Short-term use	Protracted febrile myalgia	1. Conditions for which colchicine is not effective
	Acute FMF attacks	
	Colchicine intoxication	2. Conditions for which colchicine is hazardous
	Acute kidney injury (severe)	
	Acute liver failure/injury	
	Hunger strike, voluntary fasting	3. Conditions for which oral administration of colchicine is inhibited
	Severe diarrhea for any reason	
	Abdominal operation	
	Transient impairment of gastrointestinal function for any reason	
Safety	Kidney transplantation	4. Conditions in which IL-1 blocker impose additional risk
	Dialysis	
	Recurrent infections	
	Immunocompromised patient	
	Pregnancy	
	Old age	
	Malignancy	

### 7.11. Anakinra for Pediatric Patients

Anakinra has been used and reported as a treatment for children with colchicine-resistant FMF since 2008 [164]. Dosing is adopted from treatment of pediatric CAPS (1 to 2 mg/kg/day with possible slow increments, if required, to a maximal dose of 8 mg/kg). Published case series on anakinra treatment in children with colchicine-resistant FMF report on remission and reduction of attack rate and levels of inflammatory markers, similarly to other reports for adults [165,166,167]. This success led to increased attendance in school. A major concern is the poor compliance, due to the reluctance of children and parents to undergo subcutaneous injections on a daily basis. Indeed, the studies report on a large proportion of children, who switched to canakinumab due to this distress. Other reasons for treatment change being side effects or treatment failure in rates comparable to those in adults. 

### 7.12. Anakinra in Conception, Pregnancy and Lactation

Colchicine is considered safe in pregnancy [168]. Therefore, we recommend continuing its use throughout pregnancy, and even, if required, to raise its dose to control increased FMF activity, which may occur in a third of FMF pregnancies [169]. Regarding the safety of anakinra and canakinumab during pregnancy, an opinion is still being conceived. Data are encouraging, though scant, and suggest that anakinra is safe both to mothers and fetuses [170,171]. 

In one series, 21 of 23 anakinra-exposed pregnancies resulted in healthy infants and one in miscarriage. One infant was born with unilateral renal agenesis and ectopic neurohypophysis. The population was mixed, mainly consisting of CAPS patients; only three had FMF. Exposure to anakinra during the whole course of pregnancy featured 40% of patients. None of the mothers or infants suffered of infection. Ten of twenty-two babies were breast fed by mothers taking anakinra, with no reported infections or developmental abnormalities [172]. Another study with nine pregnancies in CAPS patients, confirmed these results [173]. Of note is that the low incidence of adverse pregnancy outcomes may be biased by the better monitoring and control of inflammation offered to treated compared to untreated patients, and suppression of IL-1β, which may underlie abortions, as it was found to be upregulated in the decidua of women with recurrent pregnancy loss [174,175]. Data on babies born to men exposed to anakinra during conception is even more meager; There were no congenital or developmental abnormalities during a follow-up of four weeks to eight years in six babies born to such fathers [172]. 

At present, we advise FMF patients conceiving under anakinra to continue treatment with anakinra after discussing the pros and cons associated with this advice. This is based on our own and published experience, and on EULAR recommendations, which accept the use of antirheumatic drugs before pregnancy, and during pregnancy and lactation where there are no better alternatives [176]. We prefer anakinra over canakinumab, due to increased risk of infections during pregnancy [177] and possible incidents of pregnancy complications, conditions that may require an abrupt termination of IL-1 blockers and therefore favor one with short half-life. This selection, however, is not evidence based, and therefore difficult to implement in women who are already on canakinumab. 

### 7.13. Safety of Anakinra

Since its introduction in 2002, thousands of patients have received anakinra for numerous indications and many years, with a remarkable record of safety and tolerance. Most reported adverse effects are injection-site reactions, which may require topical corticosteroids or systemic antihistamine and usually regress within a month [178], but occasionally may lead to termination of anakinra treatment. Severe skin reactions are extremely rare and may include drug reaction with eosinophilia and systemic symptoms (DRESS) [179,180]. To the best of our knowledge, no other severe skin reactions, such as toxic epidermal necrolysis or acute generalized exanthematous pustulosis, were reported. Increased rates of routine infections are reported, but in contrast to most other inflammatory cytokine inhibitors, and biological disease-modifying anti-rheumatic drugs (DMARDs), opportunistic infections, tuberculosis, and herpes zoster are extremely rare in patients on anakinra. Serious infections such as pneumonia and pyelonephritis were also rarely reported. Other adverse events include leukopenia, headache, and urticarial rash, but their relatedness to anakinra is usually difficult to assess, given the retrospective nature of most such reports [179,180]. Malignancies were not reported in any of the studies on patients treated with anakinra. 

From the perspective of safety, anakinra’s short half-life is an advantage, particularly with respect to infection. Thus, in patients with increased risk of infections due to age-, drug-, or disease-related compromised immune response or in pregnancy, short-acting agents are preferred as discussed previously (Section 7.1, Section 7.5 and Section 7.12) and listed in Table 2.

## 8. Conclusions

FMF is the most common monogenic auto-inflammatory disorder, characterized by recurrent attacks of serositis, leg pain, chronic inflammation and amyloidosis. Colchicine is effective in most FMF patients, but approximately 10% do not respond or are intolerant to it. This hurdle is currently answered by IL-1 blockers. 

Anakinra is a recombinant form of IL-RA, the naturally occurring inhibitor of IL-1α and IL-1β. Anakinra has been shown to be safe and effective in FMF in a randomized controlled study, as well as in ample case reports. Anakinra induces remission or significantly reduces FMF activity in most patients, with an acceptable safety toll, mostly injection-site reactions in the minority of the patients. In addition to prevention of serositis attacks and ceasing inflammation, anakinra diminishes exertional musculoskeletal symptoms, a manifestation less amenable to colchicine, and abates protracted febrile myalgia, thereby preventing long-term exposure to high-dose corticosteroids. When given on time, it prevents amyloidosis-related renal function decline and may reverse proteinuria. It also appears to be safe and effective during childhood and in pregnancy, although to assure safety in pregnancy more data is required. It is effective as well as safe in FMF conditions in which there is an increased risk of acquiring infections, such as chronic dialysis, and post transplantation. 

Most importantly, due to its short half-life, anakinra emerges, mostly based on logical inference, and our own experience as the preferred IL-1 blocker in FMF and comorbidities with increased risk to acquire infection such as chronic dialysis, and post transplantation. It is the preferred IL-1 blocker also in scenarios where the continuous administration of colchicine is harmful or hindered and must be transiently terminated, such as during colchicine intoxication or an abdominal operation. It should be also preferentially used during pregnancy. Altogether, anakinra has become an essential component in FMF therapy, and the preferred IL-1 blocker under certain circumstances in FMF.

## Figures and Tables

**Table 1 ijms-23-03956-t001:** Conditions for which anakinra is useful in FMF.

Condition	Definition of the Condition	Evidence **
Colchicine resistant FMF	Failure of maximal tolerable colchicine dose to reduce FMF attack frequency to less than 1 per month	1, 2, 3
Unresolved leg pain or arthritis	Exertional leg pain, or episodic arthritis, unaffected by treatment, in otherwise colchicine responsive FMF patients	1, 2, 3
FMF-amyloidosis	Biopsy proven (at any site) AA amyloidosis, or proteinuria, highly suspected to reflect AA amyloidosis	2
Renal transplantation (amyloidosis related)	Pre- and post-kidney transplantation, performed for amyloidosis related end stage kidney disease	0 for graft outcome, 2 for amyloidosis
Protracted febrile myalgia	Protracted (≥3 weeks) of severe and debilitating myalgia, usually accompanied by fever and markedly increased inflammatory markers	2
Pericarditis	Episodic or chronic inflammation of the pericard, manifested with pleuritic chest pain, pericardial effusion and/ or typical electrocardiogram features	3
On demand treatment for acute attacks	Administration of anakinra at the onset or prodrome of an attack	2, 3
Temporary replacement of colchicine	Switching from colchicine to anakinra for a limited time in certain conditions listed in category 2 and 3 of Table 2 (e.g., colchicine intoxication)	3
Permanent replacement of colchicine	Continuous anakinra instead of colchicine, due to colchicine intolerance or allergy	3
Pediatric FMF	Treating children, mostly with colchicine refractory FMF, with anakinra at a dose of 1–2 mg/kg/day with slow increment to 8 mg/kg if required	2
Conception, pregnancy and lactation in FMF	Carry on with anakinra or switching from canakinumab to anakinra during pregnancy or lactation	2, 3

** 0, There is no supportive evidence; 1, randomized controlled trials; 2, case reports or series; 3, own experience.

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
