# Peer review of "The Preferential Use of Anakinra in Various Settings of FMF: A Review Applied to an Updated Treatment-Related Perspective of the Disease"

_ijms, 2022, doi:10.3390/ijms23073956_

Round 1

Reviewer 1 Report

The authors present an interesting and well-balanced review on the preferential use of anakinra in various settings of FMF. The manuscript is well written and the table appropriately complement the text.

This is really a nice piece of work and there is not too much I would add, however, let me suggest to expand on  non-MEFV genetic modifying factors such as SAA1, TNF -alpha and MICA either within paragraph 3.2.1. or within a separate paragraph.

Reviewer 2 Report

The paper by Livneh et al entitled: “The preferential use of anakinra in various settings of FMF - A review applied to an updated treatment-related perspective of the disease” is complex, but interesting to read, providing also a detailed description of FMF. Many abbreviations/initials used in the text are never repeated throughout the whole paper (are they uiseful?). Tables 1 and 2 appear important for the reader of IJMS.

The first reference by Chae et al has been used to express the second sentence of the paper, though it refers too quickly to the complex FMF inheritance pattern. The FMF constitutional signs have not been only described in the (very old!) paper by Sohar E et al, as the medical literature is really abundant about this topic.

In the paragraph 3.3.2 omit the comma after the abbreviation PFM.

The paragraph 4.2 dedicated to canakinumab is someway 'outside the objective of this paper' and could be downsized.

Please, at page 10, when the authors state: “IL-1β is a potent pro-inflammatory cytokine, which upregulates the expression of many other inflammatory cytokines, including IL-1α”, they should reference the statement referred to IL-1α.

The beneficial effects of anakinra have been firstly shown in non-FMF autoinflammatory diseases: this is clear. However, the paragraph 6.2 is too compact and some concepts have been reported too swiftly. Some further papers should be included among references, as anakinra had been used in cryopyrin-associated periodic syndrome with unexpected spectacular success. In addition, as anakinra is the main focus of the paper, I would suggest to emphasize its success even at a neurological level in the most severe cases of cryopyrin-associated periodic syndrome. See: Hawkins et al. Spectrum of clinical features in Muckle-Wells syndrome and response to anakinra. Arthritis Rheum 2004;50:607-12; Rigante et al. Hydrocephalus in CINCA syndrome treated with anakinra. Childs Nerv Syst. 2006;22:334-7; Marchica et al. Resolution of unilateral sensorineural hearing loss in a pediatric patient with a severe phenotype of Muckle-Wells syndrome treated with anakinra. J Otolaryngol Head Neck Surg 2018;47:9). In general terms this clarification can be crucial to underscore the overall safety of anakinra in pediatric patients.

I would suggest to rephrase the section dedicated to “Transient and permanent replacement of colchicine by anakinra”, as some concepts seem contradictory and unclear to the naive reader (for instance, anakinra use in pregnancy).

In the paragraph 7.13 I would change the adjective “ample” with “numerous”. In the last sentence of the paper, please, change “blockers” with “blocker”.

The paper shows us -through a complex entwined itinerary- that anakinra 'short' half-life can be an advantage for some patients with FMF.
